# Genome-Wide Investigation of the *NF-X1* Gene Family in *Populus trichocarpa* Expression Profiles during Development and Stress

**DOI:** 10.3390/ijms22094664

**Published:** 2021-04-28

**Authors:** Fang He, Yu-Jie Shi, Jia-Xuan Mi, Kuang-Ji Zhao, Xing-Lei Cui, Liang-Hua Chen, Han-Bo Yang, Fan Zhang, Qian Zhao, Jin-Liang Huang, Xue-Qin Wan

**Affiliations:** Sichuan Province Key Laboratory of Ecological Forestry Engineering on the Upper Reaches of the Yangtze River, College of Forestry, Sichuan Agricultural University, Chengdu 611130, China; 2019104006@stu.sicau.edu.cn (Y.-J.S.); s20177704@stu.sicau.edu.cn (J.-X.M.); zhaokj@sicau.edu.cn (K.-J.Z.); xinglei.cui@sicau.edu.cn (X.-L.C.); 41377@sicau.edu.cn (L.-H.C.); yanghanbo6@sicau.edu.cn (H.-B.Y.); 13305@sicau.edu.cn (F.Z.); 2019204003@stu.sicau.edu.cn (Q.Z.); B20162601@stu.sicau.edu.cn (J.-L.H.)

**Keywords:** genome-wide, *Populus trichocarpa*, *NF-X1* gene family, expression pattern, stress

## Abstract

Poplar are planted extensively in reforestation and afforestation. However, their successful establishment largely depends on the environmental conditions of the newly established plantation and their resistance to abiotic as well as biotic stresses. NF-X1, a widespread transcription factor in plants, plays an irreplaceable role in plant growth, development, and stress tolerance. Although the whole genome sequence of *Populus trichocarpa* has been published for a long time, little is known about the *NF-X1* genes in poplar, especially those related to drought stress, mechanical damage, insect feeding, and hormone response at the whole genome level. In this study, whole genome analysis of the poplar *NF-X1* family was performed, and 4 *PtrNF-X1* genes were identified. Then, bioinformatics analysis and qRT-PCR were applied to analyze the gene structure, phylogeny, chromosomal localization, gene replication, Cis-elements, and expression patterns of *PtrNF-X1*
*genes*. Sequence analysis revealed that one-quarter of the *PtrNF-X1* genes did not contain introns. Phylogenetic analysis revealed that all *NF-X1* genes were split into three subfamilies. The number of two pairs of segmented replication genes were detected in poplars. Cis-acting element analysis identified a large number of elements of growth and development and stress-related elements on the promoters of different *NF-X1* members. In addition, some *PtrNF-X1* could be significantly induced by polyethylene glycol (PEG) and abscisic acid (ABA), thus revealing their potential role in regulating stress response. Comprehensive analysis is helpful in selecting candidate *NF-X1* genes for the follow-up study of the biological function, and molecular genetic progress of stress resistance in forest trees provides genetic resources.

## 1. Introduction

The survival, growth, and reproduction of plants are adversely affected by a broad range of biotic and abiotic stresses [1,2]. Biotic stresses include insects, mites, harmful bacteria, herbivores, human activities, and so on [3,4]. Abiotic stresses also pose significant threats to plants, including pole-end temperatures, water deficiency or excess, high salinity, oxidative stress, ultraviolet B (UV-B) light, and chemical contaminants [5,6,7]. During the long-term evolution, plants respond to such unfavorable environmental variables via growth, developmental, cellular, molecular, physiological, and biochemical pathways [8]. In addition, osmotic stress is the main signal of abiotic stress in plants [9,10], including drought, salt, cold and heat, etc. Meanwhile, a network of transcription factors (TFs) can regulate the stress-responsive genes, whose expression control the above responses [11].

TFs account for about 7% of the coding sequence in plant genome, such as 6.2% in *Arabidopsis thaliana* and 6.4% in poplar [12]. Furthermore, TFs can interact with cis-acting elements in the promoter region of stress response genes to activate the entire gene network of signal cascade [13,14,15]. Moreover, TFs play an indispensable role in the transcription regulation of genes, which means TFs play a key part in the process of transforming stress signal sense into stress response gene expression [16,17]. To be specific, as signal molecules, hormones such as ABA, ethylene (ET), and gibberellic acid (GA) can mediate signaling pathways, which mostly rely on TFs for regulating plant responses to stress [18,19,20].

NF-X1, as one of the most important transcription factors, occurs in virtually all eukaryotes, such as *Caenorhabditis elegans*, *Schizosaccharomyces pombe*, *Drosophila*, Human, and *Arabidopsis thaliana* [21]. *NF-X1* was first discovered in 1994 and has been identified as a protein that binds to the promoter (X-Box motif) of the CLASS II MHC (major histocompatibility complex) gene, inhibiting its expression [22]. The *NF-X1* gene of *Drosophila melanogaster*, named Shuttle craft (SCT), plays a material role in embryonic development by fine-tuning the orientation or spatial maintenance of the migratory SNb and ISN nerve roots in the central nervous system (CNS) [23]. Yeast *FAP1*, a homologous gene of *NF-X1*, inhibits the cytotoxic effects of rapamycin by competing with rapamycin for binding to the peptidyl-prolyl cis-trans isomerase FKBP12. In humans, *NF-X1* can activate telomerase activity in an effort to inhibit cell senescence caused by the HPV-16 (the most carcinogenic subtype of human papillomavirus) [24,25]. More importantly, it also plays a critical role in the regulation of plant alarm reaction. AtNF-XL1 and AtNF-XL2 proteins in Arabidopsis, two members of the *NF-X1* family, are distinguished by *NF-X1* type zinc fingers, which possibly participate in DNA binding action, and PHD (plant homeodomain) finger motifs, which possibly take part in protein interactions [26]. The AtNF-XL1 protein can promote the physiological status of plants and boost their growth and survival under salt stress and may also be involved in regulatory processes that protect photosynthesis [26]. Meanwhile, *NF-X1* TFs protein in wheat regulates the differential expression of genes to resist leaf rust pathogen-mediated infection, among which the highest and lowest expression levels are *TaNF-X1-3* and *TaNF-X1-2*, separately [27]. NF-XL1 was demonstrated to be involved in seed longevity and seed coat permeability [28]. In general, *NF-X1* plays an indispensable role in plant growth, development, and response to stress. The genomes of model plants (such as rice, Arabidopsis, poplar, etc.) can reveal the function of plant genes [29,30]. Meanwhile, comparative genomics is used to study the *NF-X1* gene and to compare its genome structure to understand its function and expression mechanism.

Due to rapid growth, wide distribution, and high adaptability, *P.*
*trichocarpa* has been widely used as a timber stand, avenue tree, and shelterbelt. Meanwhile, *P. trichocarpa,* as a momentous model species for woody plant research with woody perennial growth habits, is endowed with relatively upper ecological, economic, and cultural significance [31,32,33]. Presently, most of the studies on *NF-X1* transcription factors are focused on humans and bacteria. There are few functional studies on *NF-X1* transcription factors for biological and abiotic stress in poplar trees. Four putative *NF-X1* genes in *P. trichocarpa* were studied from the aspects of molecular structure, phylogeny, chromosomal localization, and expression pattern (under ABA, PEG, and MeJA (Jasmine acid methyl ester) treatment). These results provide a solid foundation for further analyses of the functions and molecular evolution of *NF-X1* genes in poplar and, in particular, for improving the stress resistance of poplar by genetic manipulation.

## 2. Results

### 2.1. Identification and Analysis of NF-X1 in A. thaliana, Oryza sativa, and P. trichocarpa

The eight putative *NF-X1* genes were identified in the published *A. thaliana*, *O. sativa*, and *P. trichocarpa* reference genomes [34] and were successively named *AtNF-X1-1* to *AtNF-X1-2, OsNF-X1-1 to OsNF-X1-2,* and *PtrNF-X1-1* to *PtrNF-X1-4* according to their genome locations. In order to clearly understand the characteristics of the *NF-X1* family in poplar, we analyzed the Pfam ID, the length of the gDNA (genomic DNA), the length of the transcriptional sequence, the length of the CDS (coding sequence), the position of the conserved domain, the length of amino acid (AA), the protein molecular weight (MW), the grand average of hydropathicity (GRAVY), and the isoelectric point (PI) of these genes as well as the best homologs in *Arabidopsis thaliana* (Table 1).

It was clearly found that all the eight genes had the same Pfam ID, indicating that they contained the same conserved domain. Among the eight NF-X1 proteins, PtrNF-X1-1 had the minimum protein with 358 amino acids (AA), while the maximum was AtNF-X1-1 (1188 AA). The MW of the proteins ranged approximately from 39.77 to 130.72 kDa, the GRAVY of the proteins was approximately −0.616 (PtrNF-X1-2) ~ −0.099 (PtrNF-X1-1), and PI were 8.24 (PtrNF-X1-3) ~ 8.87 (OsNF-X1-1). OsNF-X1-1, PtrNF-X1-1, PtrNF-X1-2, and AtNF-X1-2 had high homology, while OsNF-X1-2, PtrNF-X1-3, and PtrNF-X1-4 had high homology with AtNF-X1-1. In general, the identification of molecular characteristics of NF-X1 will be helpful for the study of its specific biological functions.

### 2.2. Evolutionary and Phylogenetic Analysis of the NF-X1 Family

In order to better know the evolution and differentiation of NF-X1 family proteins among species, 138 NF-X1 amino acid sequences from 52 species were applied to construct an unrooted phylogenetic tree (Figure 1a). These sequences were divided into three groups (I, II, III). Among them, class I and II had the most members, including 134 members and accounting for 97.10% of the deduced NF-X1 protein. At the same time, NF-X1 proteins from *A. thaliana*, *O. sativa*, and *P. trichocarpa* were evenly distributed in I and II.

In order to further explore the origin and evolution of *NF-X1* genes, we searched for *NF-X1* genes from 52 species from aquatic plants to terrestrial plants (Figure 1b). Notably, the three species with the greatest number of *NF-X1* genes were *Glycine*
*max* (9), *Theobroma cacao* (8), and *Citrus sinensis* (6). *Salix purpurea*, a member of the salicaceae family, contained five NF-X1 proteins, which were adjacent to the PtrNF-X1 protein.

### 2.3. Gene Structure, Conserved Domain, and Motif

In order to explore the evolutionary relationship among different kinds of the *NF-X1* gene family more clearly, an unrooted phylogenetic tree was constructed using NF-X1 proteins from *A. thaliana*, *O. sativa*, and *P. trichocarpa* (Figure 2a), and their genes structure, protein conserved motif, and conserved structure were compared. Phylogenetic analysis showed that the NF-X1 proteins were clustered in the two branches. PtrNF-X1-3, PtrNF-X1-4, AtNF-X1-1, and OsNF-X1-2 belonged to the same class II, while PtrNF-X1-1, OsNF-X1-1, AtNF-X1-2, and PtrNF-X1-2 clustered in class I. In class II, except for PtrNF-X1-4, there was only one intron, and the other three genes had no intron. However, in class I, except for *PtrNF-X1-1* with 2 introns, the other 3 genes had 12 introns. Surprisingly, only the *PtrNF-X1-1* gene had no UTR region (Figure 2b).

In general, genes with high similarity in amino acid sequences in the same family may have similar biological functions. In addition, we identified eight conserved domains in the NF-X1 protein sequence by using CDD and SMART software, and they all shared one or more NF-X1-zinc-finger domain (Figure 2c and Appendix A), which may endow them with similar biological functions. At the same time, 10 conserved motifs were identified in the NF-X1 protein sequence by MEME software (Figure 2d). Amino acid sequence encode and Sq-Logo of the Motif are displayed in Appendix A. Among them, two motifs (3 and 7) were discovered to be related to the N-terminal domain, four motifs (9, 8, 5, 6) were related to the intermediate NF-X1-zinc-finger domain, and four motifs (10, 2, 4, 1) were related to the C-terminal R3-H domain. Moreover, PtrNF-X1-1 contained one motif (8).

### 2.4. Analysis of Chromosomal Location and Gene Duplication

To further understand the evolution and differentiation of *NF-X1* family genes, we analyzed the chromosomal distribution, synteny, and evolution of 4 *NF-X1* genes in *poplar*. The *PtrNF-X1* is distributed on chromosome 8, 10, 12, and 15 (Figure 3). The higher sequence similarity between repetitive gene pairs indicates that they may be involved in the regulation of analogical biological processes. The formation of this duplication gene pair mainly included tandem and segmental duplication events [30]. Furthermore, we found by using the MCScanX method that four genes (*Ptr**NF-X1**-1**/2 and PtrNF-X1-3/4*) formed two segmental duplication events (Appendix A). These results suggest that *PtrNF-X1* was likely to be caused by gene replication, while the segmental replication events mainly catalyzed the evolution of the *NF-X1* gene in poplar.

To further analyze the evolution of the *NF-X1* gene in the plants, we first confirmed the location of the *NF-X1* gene on the chromosomes of *A. thaliana, O. sativa,* and *P. trichocarpa* (Figure 4a) and constructed comparative syntenic maps of poplar and two typical plants (Figure 4b). According to the syntenic maps, two pairs of paralogous genes were found between *P. trichocarpa* and *A. thaliana*, and one pair of paralogous genes was found between *P. trichocarpa* and *O. sativa* (Figure 4b and Appendix A). Surprisingly, *PtrNF-X1-1* was found to be associated with synonym gene pair (especially between poplar and Arabidopsis and between poplar and rice *NF-X1* genes), suggesting that this gene might play a critical role in the evolutionary process of the *NF-X1* gene family.

### 2.5. Cis-Elements Analysis of the PtrNF-X1

To further determine the expression pattern of *PtrNF-X1*, the promoter fragment (~2000 bp) upstream of the transcription starting site was isolated from *P. trichocarpa* genomic DNA. First, the cis-elements of the *PtrNF-X1* promoter were analyzed by PlantCARE database (Figure 5 and Appendix A). A series of cis-acting elements were identified as being involved in development, including seed-specific regulation, endosperm expression, gibberellin responsiveness element, and auxin responsiveness element, and all of these indicated the crucial role of cis-acting elements in poplar developments of *PtrNF-X1*.

In addition, there were also some stress or stress-related elements in the promoter region, including MeJA responsiveness, salicylic acid responsiveness, abscisic acid responsiveness, drought inducibility, defense and stress responsiveness, anoxic specific inducibility element, low-temperature responsiveness, and anaerobic induction, which meant *PtrNF-X1* might also respond to stress. For example, all *PtrNF-X1* gene promoters contained at least two elements of abscisic acid (ABA) responsiveness (Appendix A), indicating that most of the *NF-X1* gene was involved in ABA signal transduction. In addition, *PtrNF-X1-1* had multiple elements related to adversity, including two elements of MeJA-responsiveness, two elements of salicylic acid responsiveness, three elements of abscisic acid responsiveness, and two elements of drought-inducibility (Appendix A), suggesting that *PtrNF-X1-1* may respond to both biotic and abiotic stress.

### 2.6. Expression Level of Poplar NF-X1 Genes with RNA-seq

So as to explore the role of the *NF-X1* gene in the growth and development of poplar, we analyzed the tissue expression patterns of four *PtrNF-X1* genes in the transcriptomic dataset. First, we constructed a heat map of dual clustering (sample and gene) to reveal the expression profile of these genes in 15 tissues of *poplar* (Figure 6a and Appendix A). Oddly enough, there is no good rule in the expression of four NF-X1 genes in different tissues and development of poplar. For example, through RNA-seq analysis, *PtrNF-X1-1* was significantly upregulated in bud-pre-chilling and twigs-non-girdled, but significantly downregulated in leaves-non-girdled. At the same time, *PtrNF-X1-3* was dramatically upregulated in leaves-non-girdled and leaves-mature, *PtrNF-X1-2* was upregulated in leaves-non-girdled and bud-dormant, and *PtrNF-X1-4* was significantly upregulated in cambium-phloem-dormant, compared with other tissues.

In order to further understand the role of the *PtrNF-X1* gene in poplar response to pressures, we mapped the expression configurations of four *PtrNF-X1* genes in drought, beetle-damaged, and mechanical-damage through the uploaded transcriptome dataset (Figure 6b and Appendix A). According to the heat map, *PtrNF-X1-2* in leaves was markedly induced by drought stress, and *PtrNF-X1-1* in root was markedly induced by drought stress. However, the expression of *PtrNF-X1-3/4* was significantly inhibited under drought stress and the stress of insect biting and mechanical injury. These results will be helpful for our future research on gene function.

### 2.7. Expression Profile of PtrNF-X1 Genes in Different Plant Tissues and in Response to Different Treatments

In order to verify the reliability of the tissue expression of *NF-X1* genes in the transcriptome, four *NF-X1* (*PtrNF-X1-1/2/3/4*) genes were selected for RT-qPCR detection (Figure 7a). Consistent with the previous results, *PtrNF-X1-1* and *PtrNF-X1-2* had similar tissue expression patterns. *PtrNF-X1/2* in young leaf and root were upregulated significantly. In addition, *PtrNF-X1-2* was significantly induced in phloem. At the same time, *PtrNF-X1-3/4* was significantly upregulated in adult leaf and young leaf, while *PtrNF-X1-3* was significantly downregulated in root.

To deeply validate whether the abundance of *NF-X1* genes were affected by abiotic stress and hormone treatment, four *NF-X1* genes were carefully chosen for qRT-PCR assay to analyze their expression profiles in answer to PEG, ABA, and MeJA treatment (Figure 7b–d). In general, each treatment may significantly induce and inhibit *NF-X1* genes in poplar. For PEG treatment, only *PtrNF-X1-2* was significantly induced, while *PtrNF-X1-1/3/4* was significantly inhibited. For ABA treatment, *PtrNF-X1-2* was significantly induced, while *PtrNF-X1-3*/4 was significantly inhibited. Meanwhile, *PtrNF-X1-1* was induced at 1 and 6 h, while inhibited at 9 and 12 h. Furthermore, for MeJA treatment, *PtrNF-X1-1/4* was significantly inhibited, while *PtrNF-X1-2* had no response. Simultaneously, *PtrNF-X1-3* was significantly upregulated at 1 h MeJA treatment, but significantly inhibited at 9 h.

## 3. Discussion

Biotic and abiotic stresses will directly lead to a decline in agricultural and forestry productivity, forming a critical danger to the world’s energy, biosphere balance, and security [1,35,36]. Transcription factors play an indispensable role in regulating plant growth and development and plant response to stress [29,37]. NF-X1, as a widespread transcription factor in plants, plays an irreplaceable role in plant growth, development, and stress tolerance [28]. With the development of whole genome sequencing, more and more NF-X1 TFs have been identified in virtually all eukaryotes, such as *Caenorhabditis elegans*, *Schizosaccharomyces pombe*, *Drosophila*, Human, and *Arabidopsis thaliana* [21]. However, the NF-X1 family in the woody model plant poplar has not been deeply understood.

In this study, four *NF-X1* genes were identified in poplar, and the characteristics of all the products after replication, transcription, and translation of this gene family were analyzed, as well as was the complete construction of their systematic evolution and expression model. Referencing previous gene family studies [38], the *NF-X1* genes in poplars were named as *PtrNF-X1-1* to *PtrNF-X1-4* according to their chromosomal locations (Table 1 and Figure 4a). The model plants Arabidopsis thaliana, rice, and poplar were applied to the comparative genomics of the *NF-X1 family*.

With the exception of PtrNF-X1-1, all NF-X1 proteins have conserved domains (Table 1 and Figure 2c), which may endow them with similar biological functions. It has been reported that the loss of protein structure will lead to different functions of genes [30]. Since PtrNF-X1-1 has only one NF-X1-zinc-finger domain (Figure 2c), the length of gDNA and the number of motifs in the protein are significantly lower than other genes (Figure 2b,d). In class I, except for *PtrNF-X1-4*, which has only one intron, the other three genes had no intron (Figure 2b). The research has demonstrated that introns play necessary roles in the regulation of gene transcriptome [39]. In order to respond quickly to stress, organisms themselves need to stimulate genes quickly, and the gene structure with few or no introns will contribute to the formation of mRNA [40]. In many plants, *NF-X1* genes respond quickly to stress [27].

In order to further determine the origin and evolution of *NF-X1* genes, we constructed an evolutionary tree from 52 plants and divided it into three sections. Concordant with the above results (Figure 2a), PtrNF-X1 protein was found in group I and II (Figure 1a). PtrNF-X1-2 and AtNF-X1-2 are highly similar in gene structure, conserved protein structure, and motif (Figure 2), so they have high homology (Figure 1), revealing that sequence homology is directly related to biological function. The phylogenetic tree analysis of family genes can clearly describe the evolutionary process of genes [41]. Because poplar and willow belong to the same department [38,42], the five NF-X1 proteins from *Salix purpurea* were adjacent to the *Ptr*NF-X1 proteins. Surprisingly, only two genes were found in *A.*
*thaliana* and *O. sativa,* indicating that the number of *NF-X1* was independent from the genome size of the species (Figure 1b). Notably, the three species with the highest distribution of *NF-X1* were Glycine max (9), Theobroma cacao (8), and Citrus sinensis (6). Concurrently, two gene pairs were found between Arabidopsis and poplar, while one gene pair was found between rice and poplar (Figure 4b), indicating that the differentiation and replication of the *NF-X1* gene in poplar may be later than that in Arabidopsis and rice.

The increase of gene family members and the mechanism of genome evolution mainly depend on gene replication events, including tandem replication and segmented replication [43]. In this research, a total of two pairs of segmented replication genes was found in poplar, indicating that segmental repeats mainly contribute to the evolution of *PtrNF-X1* genes in *poplar* (Figure 3). Previous gene family studies have shown that segmental repeats genes may have similar functions and expression patterns [39,44]. For example, *PtrNF-X1-3/4* shows a downward trend under the threat of adversity (Figure 6). Similar expression levels indicate the similar function and structure of segmented replication *NF-X1* in *poplar*.

The expression pattern of the *NF-X1* gene in diverse tissues has been depicted in numerous species [26,28,45]. Due to the difference in the number of NF-X1 contained in different species, there is no uniform gene expression profile of the *NF-X1* gene in plants. According to the RNA-seq data of poplar, *PtrNF-X1-3* preferentially expressed in leaves and PtrNF-X1-4 was significantly upregulated in cambium-phloem-dormant (Figure 6a). Consistent with the RNA-seq data, fluorescence quantitative PCR results also showed that the *PtrNF-X1-4* was expressed preferentially in phloem compared with other tissues (Figure 7a). In summary, there is no obvious rule in the expression of four *NF-X1* genes in different tissues and development of poplar.

The expression level of the *NF-X1* gene in plants is closely related to stress and mainly responds to salt stress and biotic defense [26,28]. Through transcriptome dataset analysis, some *NF-X1* genes in poplar were significantly induced (*PtrNF-X1-1/2*) and inhibited (*PtrNF-X1-3/4)* by drought. A similar trend was obtained by RT-qPCR, and ABA treatments significantly upregulated the expression of *PtrNF-X1-1/2* (Figure 7c). Through analyzing the promoters of these two genes, we found that there were multiple elements of abscisic acid responsiveness in this promoter (Figure 5 and Appendix A). Moreover, many excellent reviews have indicated that the ABA signaling pathway is the core pathway for regulating plant response to stress [11,46]. Similarly, ABA signaling is involved in stress responses to drought stress, beetle damage, and mechanical damage in poplars (Figure 7c). In summary, the expression level of *NF-X1* under PEG and ABA treatment was selected to reflect the transcription level of *NF-X1* in poplar drought stress, beetle damage, and mechanical damage.

In addition, NF-X1 TFs protein in wheat regulates the different expression of genes for resisting leaf rust pathogen-mediated infection [27]. Furthermore, transcriptome verified that *PtrNF-X1-1/4* was significantly inhibited under damage and mechanical damage of poplar beetle (Figure 6b). MeJA is involved in the signal transduction of insect bites and mechanical damage to plants [47,48]. Moreover, elements of MeJA-responsiveness were identified on the promoter of *PtrNF-X1-1/3*. Simultaneously, RT-qPCR detected that *PtrNF-X1-3* was also significantly induced by MeJA, while *PtrNF-X1-1/4* was significantly inhibited under MeJA treatment (Figure 7d). Furthermore, *PtrNF-X1-*3 and *PtrNF-X1-*4 have similar gene structures and protein domains (Figure 2b,c), and are clustered into class II (Figure 1), giving them similar expression patterns (under PEG, ABA, and MeJA treatment) (Figure 7). Compared with the expression profiles described by RNA-seq data, the patterns of qRT-PCR expression are not exactly the same. The differences in expression may be caused by a variety of reasons. In general, the discoveries afford perceptions to the latent functional role of *PtrNF-X1* genes in poplar. Comprehensive analysis is useful for selecting candidate *PtrNF-X1* genes for further functional description, and genetic modification of stress resistance in forest trees provides genetic resources.

## 4. Materials and Methods

### 4.1. Retrieval of NF-X1 Genes in A. thaliana, O. sativa, and P. trichocarpa

The extraction and identification of poplar family members referred to the published papers [49]. The genome data of *A. thaliana*
*(Athaliana_447_TAIR10.fa.gz)*, *O. sativa*
*(Osativa_323_v7.0.fa.gz),* and *P. trichocarpa*
*(Ptrichocarpa_533_v4.0.fa.gz)* were obtained from the Phytozome database (https://genome.jgi.doe.gov/portal/pages, accessed on 16 January 2021). In addition, BLASTP and HMMER software were used to identify 8 NF-X1 proteins with conserved structures in Arabidopsis (2), rice (2), and poplar (4). The redundant sequences were manually discarded. Furthermore, the conservative structure, molecular weights, isoelectric points, and hydrophilicity analysis of the identified NF-X1 were carried out by SMART, ExPASy server, InterProand, and Protein GRAVY software (https://web.expasy.org/protparam/, accessed on 16 January 2021).

### 4.2. Evolutionary Relationships of NF-X1 Genes

A total of 138 NF-X1 proteins from *Amaranthus hypochondriacus, Amborella trichopoda, Ananas comosus, Aquilegia coerulea, Arabidopsis thaliana, Boechera stricta, Brachypodium distachyon, Brachypodium stacei, Capsella rubella, Carica papaya, Chlamydomonas reinhardtii, Citrus clementina, Citrus sinensis, Coccomyxa subellipsoidea, Daucus carota, Eucalyptus grandis, Eutrema salsugineum, Fragaria vesca, Glycine max, Gossypium raimondii, Linum usitatissimum, Malus domestica, Manihot esculenta, Marchantia polymorpha, Medicago truncatula, Mimulus guttatus, Musa acuminata, Oropetium thomaeum, Oryza sativa subsp. Japonica, Panicum hallii, Panicum virgatum, Phaseolus vulgaris, Physcomitrella patens, Populus euphratica, Populus trichocarpa, Prunus persica, Salix purpurea, Setaria italica, Setaria viridis, Sisymbrium irio, Solanum lycopersicum, Solanum tuberosum, Sorghum bicolor, Sphagnum fallax, Spirodela polyrhiza, Theobroma cacao, Trifolium pratense, Triticum aestivum, Vitis vinifera, Volvox carteri, Zea mays, and Zostera marina* were obtained from the Phytozome database [34]. They all have NF-X1-zinc-finger conserved domains. The amino acids of all NF-X1 target sequences were analyzed by ClustalX and then the phylogenetic tree was constructed by the neighbor-joining (NJ) method using MEGA7.0 software. The accession numbers and names of the genes are all shown in Appendix A.

### 4.3. Analysis of Gene Structure, Protein Structure, and Motifs

As these three model plants can all be searched for well-patched and complete genomic data [30,38], rice, Arabidopsis, and *Populus trichocarpa* were selected for the study of gene structure, protein structure, and motifs. The Gene Structure Display Server (http://gsds.cbi.pku.edu.cn/, accessed on 16 January 2021) was used to analyze the introns, exons, and UTR regions of the NF-X1 gene in Arabidopsis (2), rice (2), and poplar (4). The conserved motifs and domains of candidate NF-X1 proteins were identified by MEME (https://meme-suite.org/meme/tools/meme, accessed on 16 January 2021) program and CDD database (https://www.ncbi.nlm.nih.gov/Structure/bwrpsb/bwrpsb.cgi?, accessed on 16 January 2021), respectively. Finally, the conserved motifs and structures of all NF-X1 proteins in poplar were drawn by TBtools software (https://github.com/CJ-Chen/TBtools, accessed on 16 January 2021). The specific parameter setting that was used is referred to in the published papers [50].

### 4.4. Chromosomal Locations and Gene Duplication

The chromosomal positions of *NF-X1* were collected from the Phytozome database [34]. The MCScanX software was applied to analyze *NF-X1* gene duplication events [51]. The TBtools software was used to display the locations and the collinearity of *NF-X1* genes.

### 4.5. Analysis of Cis-Regulatory Elements

The promoter sequences (2000 bp upstream from the start codon) of *NF-X1* were analyzed online by Plant-CARE (http://bioinformatics.psb.ugent.be/webtools/plantcare/html/, accessed on 16 January 2021) database [52], and all cis-regulatory elements related to hormones and stress were identified.

### 4.6. Transcriptomic Data Sets to Analyze the Expression Patterns of PtrNF-X1

In order to discuss gene expression profiles of *PtrNF-X1*, the publicly available transcriptomic data were obtained from the popgenie (https://popgenie.org/, accessed on 16 January 2021) database [53]. In this study, transcriptome data were collected from stress (drought, mechanical damage, insect beetle damage) and 15 different tissues (non-girdled twigs, dormant flowers, expanded flowers, mature leaves, expanding flowers, whole-sucker suckers, mature petiole, prechilling buds, dormant buds, freshly expanded leaves, non-girdled leaves, girdled leaves, mature seeds, young expanding leaves, and dormant cambium phloem) during the growth and development of *P. trichocarpa.* The relative expression of *PtrNF-X1* was displayed as a heat map generated from TBtools software [50].

### 4.7. Plant Materials and Treatments

One-year-old *Populus trichocarpa* seedlings were planted in a plastic plot (16.0 h light; 20–25 °C; 70% air humidity at Wenjiang, Chengdu, China (30°70 N, 103°85 E, 537.11 m above sea level)). Because of its clear and complete genome, *Populus trichocarpa* was used as a woody model plant to study the gene function of poplar [54]. The previously published papers have shown that PEG and ABA treatments have been used to investigate gene responses to abiotic stresses in plants [55,56,57,58]. The 60-day-old seedlings were treated with PEG, ABA, and MeJA. There were at least 5 biological repeats in the gradient of each treatment. For each treatment, please refer to previously published papers with appropriate revisions [29,30]. For PEG treatments, similarly grown seedlings of *P. trichocarpa* (40–50 cm high, with 30–35 leaves) were subjected to a 15% PEG6000 solution for 0, 1, 3, 6, 9, and 12 h. For phytohormone analysis, similarly grown seedlings of *P. trichocarpa* were treated with a solution containing 100 μM ABA (Sigma, Santa Clara, CA, USA) and MeJA (Sigma, Santa Clara, CA, USA), respectively, for 0, 1, 3, 6, 9, and 12 h. At the time point of various treatments, the leaves were separated from the plant, rapidly frozen in liquid nitrogen, and deposited in an ultra-low temperature freezer (Thermo, Waltham, MA, USA).

For the sampling of poplar tissues and organs, please refer to a previously published paper [29]. In the absence of any stress, plant tissues and organs were sampled, and each plant tissue and organ had at least 5 biological replicates. Finally, we simultaneously collected different organs and tissues of 2-month-old *P. trichocarpa*, including young leaves, mature leaves, old leaves, xylem, phloem, and roots and immediately immersed them in liquid nitrogen.

### 4.8. RNA Extraction and Quantitative Real-Time (qRT-PCR) Analysis

Total RNA was prepared using the plant total RNA extraction kit (Aidlab, Beijing, China) according to the manufacturer’s protocol, and cDNA was reverse-transcribed from 2 μg of total RNA with the Tiangen Fast Quant RT Kit (Tiangen Biotech Co. Ltd., Beijing, China). For qRT-PCR, we refer to the previous experimental operation [35]. According to the target gene fragment, Primer Premier 6.0 was used to design primers online, and all primers are presented in Appendix A. At least 20 replicates per experiment (5 biological reproductions ×4 technical reproductions) were performed.

## 5. Conclusions

Herein, whole genome analysis of the poplar *NF-X1* family was accomplished, and four *PtrNF-X1* genes were identified. Then, bioinformatics and qRT-PCR were applied to analyze the gene structure, phylogeny, chromosomal localization, gene replication, cis-elements, and expression patterns of *PtrNF-X1*. Finally, we found that *NF-X1* genes were expressed preferably in leaf and root, and some genes could be significantly induced by PEG, ABA, and MeJA, showing that this gene (*PtrNF-X1-1/2/3*) plays a crucial part in the stress resistance of poplars. This research will contribute to the further exploration of the function of *NF-X1* genes in *poplar*.

## Figures and Tables

**Figure 1 ijms-22-04664-f001:**
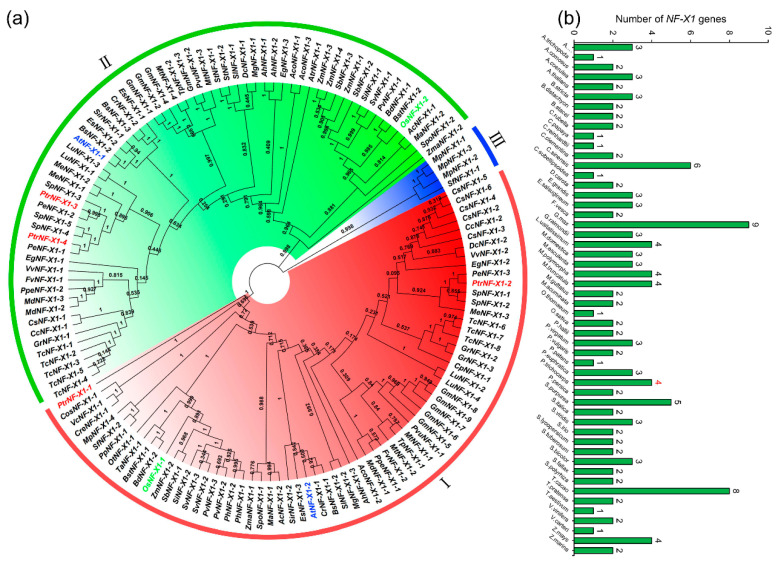
Evolutionary and phylogenetic analysis of the NF-X1 family in diverse plant species. (**a**) The phylogenetic tree of NF-X1 proteins in 52 plant species. The arcs of different colors and I, II, III represent different classes of NF-X1 domains. The red, green and blue characters represent the NF-X1 domains from poplar, rice and Arabidopsis, respectively. (**b**) Comparisons of NF-X1 proteins number across 52 species of plant.

**Figure 2 ijms-22-04664-f002:**
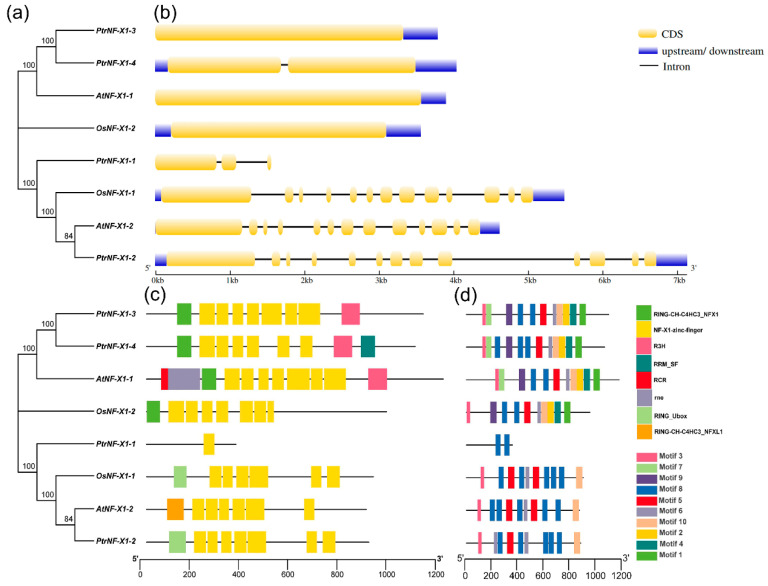
Phylogenetic, gene structure, conserved protein structure, and motif analyses of *NF-X1* gene from *A. thaliana*, *O. sativa*, and *P. trichocarpa*. (**a**) Phylogenetic tree composed of total length amino acid sequences from NF-X1 proteins. (**b**) Exon–intron structure analysis of *NF-X1* genes. Blue boxes represent UTR regions; yellow boxes indicate exons; black lines represent introns. (**c**) Conserved protein domain analysis of NF-X1 proteins. (**d**) Motif distribution of NF-X1 members. The length of each pattern is displayed proportionally.

**Figure 3 ijms-22-04664-f003:**
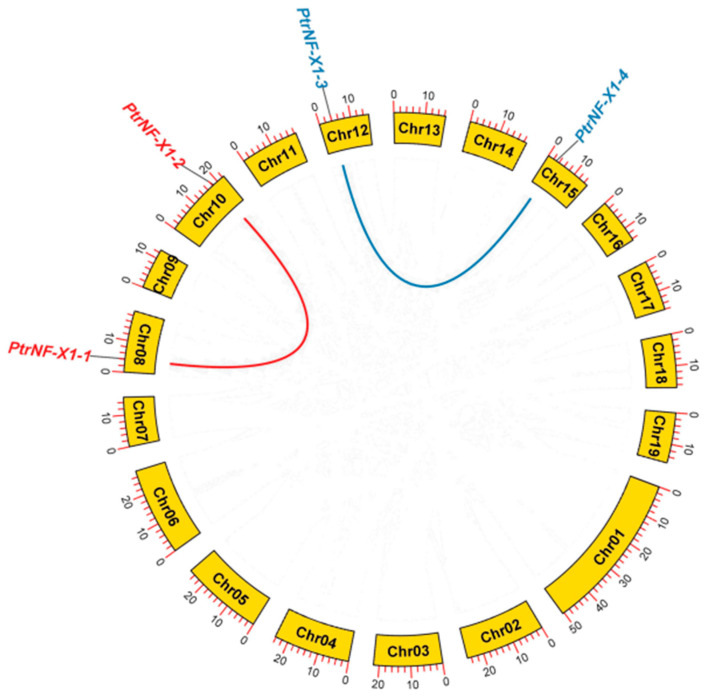
Schematic diagram of *NF-X1* gene distribution and inter-chromosomal relationships in *P. trichocarpa* chromosomes. The gray lines represent all synteny blocks in the chromosome of *P.*
*trichocarpa*. Red and blue lines show duplicated *NF-X1* gene pairs in *P.*
*trichocarpa*. The chromosome number is shown in the middle of the arc square. The length of each arc corresponds to the length of the chromosome (Mb).

**Figure 4 ijms-22-04664-f004:**
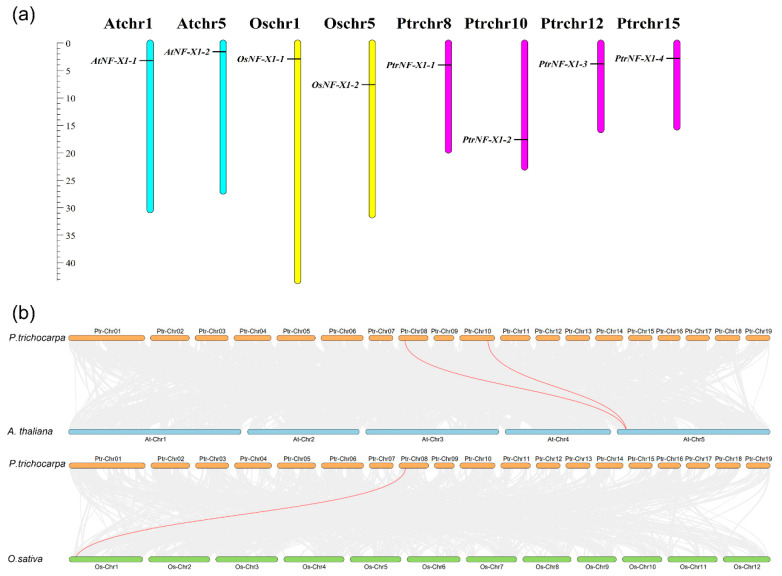
Synteny analysis of *NF-X1* genes. (**a**) Locations of NF-X1 family genes on the chromosomes of *A. thaliana, O. sativa,* and *P. trichocarpa*. (**b**) Synteny analysis of *NF-X1* genes between *poplar* and the other two model plants. Gray lines in the background represent the co-linear blocks in *poplar* and other model plant genomes, while the red lines emphasize the syntenic *NF-X1* gene pairs.

**Figure 5 ijms-22-04664-f005:**
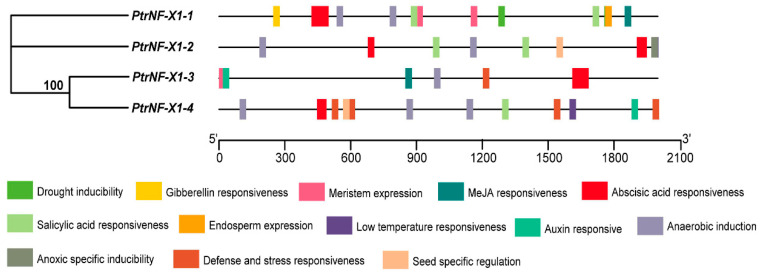
Cis-elements analysis of *PtrNF-X1*. The relative locations of stress and growth related cis-elements in *PtrNF-X1* promoter region. Different colors represent different cis-acting elements, and their position corresponds to the corresponding position of the promoter.

**Figure 6 ijms-22-04664-f006:**
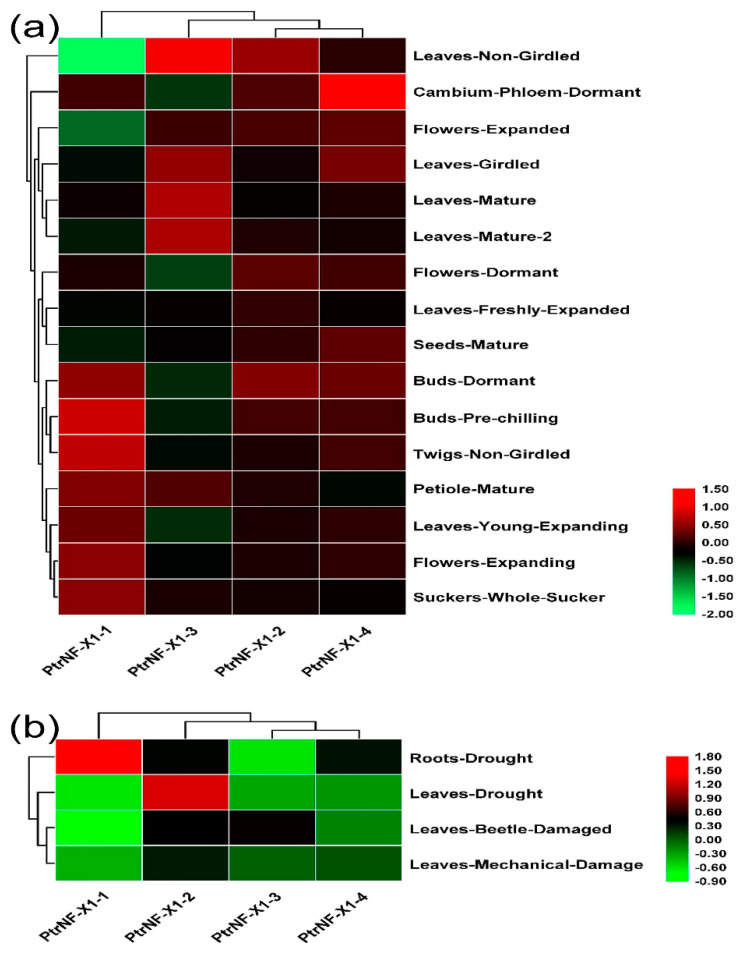
Expression profile of *PtrNF-X1* gene under development and stress. (**a**) Heat maps showing the expression levels of four *PtrNF-X1* gene in different tissues at different developmental stages based on transcriptome data. (**b**) Heat maps of the expression levels of four *PtrNF-X1* genes in drought, beetle damage, and mechanical damage. The color bar represents the range of maximum and minimum values of relative expressions in the heat map. Value = log_2_ (fold change).

**Figure 7 ijms-22-04664-f007:**
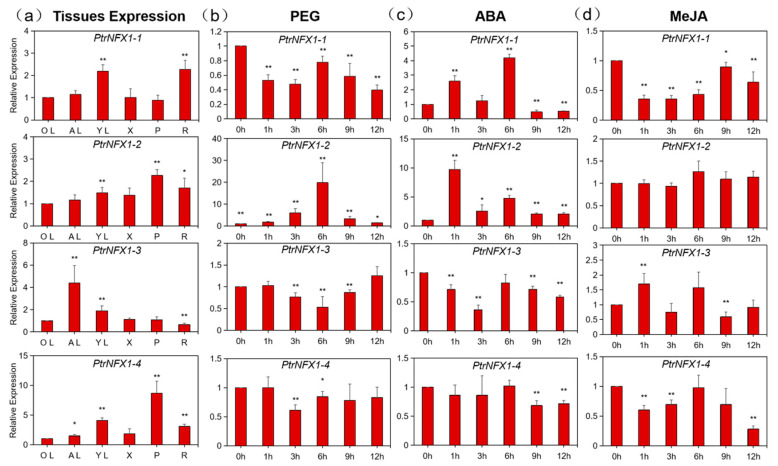
RT-qPCR analysis of the expression profile of *PtrNF-X1* gene. (**a**) Transcript-level analysis of *PtrNF-X1* in various tissues. YL, young leaf; AL, adult leaf; OL, old leaf; P, phloem; X, xylem; R, root. Transcript levels of *PtrNF-X1* were measured by RT-qPCR in response to PEG treatment (**b**), ABA treatment (**c**), and MeJA treatment (**d**). Values are means + SE (*n* = 20). Asterisks represent significant differences: * *p* ≤ 0.05; ** *p* ≤ 0.01.

**Table 1 ijms-22-04664-t001:** Summary of NF-X1 in *A. thaliana*, *O. sativa*, and *P. trichocarpa*. AA, amino acid; MW, molecular weight (kDa); GRAVY: grand average of hydropathicity; PI: theoretical isoelectric point.

Name	Gene Model ID	Description	Pfam ID	gDNA	Transcript	CDS	Domains	AA	MW	GRAVY	PI	Homologs in Arabidopsis
*AtNF-X1-1*	AT1G10170.1	NF-X1 family protein	PF01422	3897	3897	3567	58-134/86-215/222-279/312-373/380-428/448-489/503-550/562-622/601-653/658-705/711-766/749-798/888-963	1188	130.72	−0.601	8.54	/
*AtNF-X1-2*	AT5G05660.1	NF-X1 family protein	PF01422	4617	2908	2643	82-149/183-231/237-285/290-330/344-393/397-445/424-472/631-672	880	98.88	−0.605	8.73	/
*OsNF-X1-1*	LOC_Os01g06550.1	NF-X1 family protein	PF01422	5482	3225	2730	109-160/252-300/305-345/359-408/412-460/439-487/659-700/723-774	910	99.56	−0.432	8.87	AT5G05660.1(*AtNF-X1-2*)
*OsNF-X1-2*	LOC_Os06g14190.1	NF-X1 family protein	PF01422	3559	3559	2886	1-54/87-150/157-205/221-264/279-327/374-426/431-479/484-510	962	103.89	−0.346	8.34	AT1G10170.1(*AtNF-X1-1*)
*PtrNF-X1-1*	Potri.008G068500.1	NF-X1 family protein	PF01422	1557	1077	1077	228-272	358	39.77	−0.099	8.67	AT5G05660.1(*AtNF-X1-2*)
*PtrNF-X1-2*	Potri.010G188700.1	NF-X1 family protein	PF01422	7128	3237	2673	90-157/190-238/244-284/299-340/351-400/404-452/431-479/641-682/705-756	891	99.75	−0.616	8.81	AT5G05660.1(*AtNF-X1-2*)
*PtrNF-X1-3*	Potri.012G043700.1	NF-X1 family protein	PF01422/PF01424	3786	3786	3327	122-179/212-273/280-328/344-387/402-450/460-519/498-541/555-603/608-663/646-695/781-854	1109	119.81	−0.429	8.24	AT1G10170.1(*AtNF-X1-1*)
*PtrNF-X1-4*	Potri.015G034500.1	NF-X1 family protein	PF01422	4035	3920	3231	122-179/212-273/280-328344-387/402-450/524-572/615-664/750-823/858-915	1077	116.40	−0.375	8.55	AT1G10170.1(*AtNF-X1-1*)

## Data Availability

The data presented in this study are available in a Appendix A.

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
