# Peer review of "Genome-Wide Investigation of the NF-X1 Gene Family in Populus trichocarpa Expression Profiles during Development and Stress"

_ijms, 2021, doi:10.3390/ijms22094664_

Round 1

Reviewer 1 Report

Dear Colleagues,

Thank you for rewriting the manuscript. Now the text is more correct and easy-to read as was it before.

Author Response

Dear Editor and reviewer:

We would like to thank you and the reviewers for your valuable comments and suggestions that helped us further our understanding of some aspects on this field and to improve our manuscript. After carefully considering the Reviewers’ comments, we revised our manuscript by point-by-point, and hereby list our responses to these comments. On the basis of the previous version, we provided a trace version and re-uploaded it.

Please do not hesitate to contact me in case the manuscript has any further issues.

Thank you very much for your help.

We are looking forward to hearing from you soon.

Yours sincerely,

Fang He, Ph.D.

Associate professor of forest tree genetics and breeding

College of Forestry,

Sichuan Agricultural University , Chengdou 611130 , China

Tel: +86-17683777884

E-mail: [email protected][email protected]

Reviewer 2 Report

The paper is interesting and well write. 

The aim of study have to be improved. In the current version it is not claer!

Author Response

manuscript ID (ijms-1201645)

Dear Editor:

We would like to thank you and the reviewers for your valuable comments and suggestions that helped us further our understanding of some aspects on this field and to improve our manuscript. After carefully considering the Reviewers’ comments, we revised our manuscript by point-by-point, and hereby list our responses to these comments. On the basis of the previous version, we provided a trace version and re-uploaded it.

Please do not hesitate to contact me in case the manuscript has any further issues.

Thank you very much for your help.

We are looking forward to hearing from you soon.

Yours sincerely,

Fang He, Ph.D.

Associate professor of forest tree genetics and breeding

College of Forestry,

Sichuan Agricultural University , Chengdou 611130 , China

Tel: +86-17683777884

E-mail: [email protected][email protected]

-------------------------------------------------------------------------------------------------------

Responses to reviewer 2:

【Comment 1】L11"Poplar are planted extensively ...."

L32 Start with "The survival, growth ......"

L263 what does GW have to do with it?. Start with "Biotic and abiotic..."

[Our response1]: Thank you for your advice. We have revised this sentence in the original text.

The abstract has been updated in this paper.

The original is “Trees such as poplar … and afforestation.”, and we have revised it to “Poplar are planted extensively in reforestation and afforestation.” (in line 11).

The original is “As it is well known, the survival…and abiotic stresses”, and we have revised it to “The survival, growth, and reproduction of plants are adversely affected by a broad range of biotic and abiotic stresses” (in line 32-33).

The original is “Global warming will cause serious harm…biosphere balance and security”, and we have revised it to “Biotic and abiotic stresses will directly lead to a decline in agricultural and forestry productivity, forming a critical danger to the world's energy, biosphere balance and security [1-3].” (in line 263-266).

【Comment 2】L84repharese! the aim is not clear!

[Our response 2]: Thank you for your question. We rewrote the sentence to make the purpose of the article clearer.

The Introduction has been updated in this paper.

The original is “Four putative NF-X1 genes in P. trichocarpa were… to cultivate poplar with high resistance to adversity.”, and we have revised it to “Four putative NF-X1 genes in P. trichocarpa were studied from the aspects of molecular structure, phylogeny, chromosomal localization, and expression pattern (under ABA, PEG, and MeJA treatment). These results provide a solid foundation for further analyses of the functions and molecular evolution of NF-X1 genes in poplar, and, in particular, for improving the stress resistance of poplar by genetics manipulation.” (in line 85-90).

【Comment 3】L477 which gene?  

[Our response 3]: Thank you for your question. To make the results clearer, we have added the names of these genes.

As nicely advised, the Conclusions have been updated in this paper.

The original is “Finally, we found that NF-X1 genes were … resistance of poplars.”, and we have revised it to “Finally, we found that NF-X1 genes were expressed preferably in leaf and root, and some genes could be significantly induced by PEG, ABA and MeJA, showing that this gene (PtrNF-X1-1/2/3) plays a crucial part in the stress resistance of poplars.” (in line 448-451).

Thank you again for your questions and comments, which have been revised in the original text.

  1. He, F.; Niu, M. X.; Feng, C. H.; Li, H. G.; Su, Y.; Su, W. L.; Pang, H.; Yang, Y.; Yu, X.; Wang, H. L.; Wang, J.; Liu, C.; Yin, W.; Xia, X., PeSTZ1 confers salt stress tolerance by scavenging the accumulation of ROS through regulating the expression of PeZAT12 and PeAPX2 in Populus. Tree Physiol 2020.
  2. Zhou, J. M.; Zhang, Y., Plant Immunity: Danger Perception and Signaling. Cell 2020,181, (5), 978-989.
  3. Zhao, C.; Zhang, H.; Song, C.; Zhu, J.-K.; Shabala, S., Mechanisms of Plant Responses and Adaptation to Soil Salinity. The Innovation 2020,1, (1), 100017.
  4. Zhimin Song, S. K., $ Dimitris Thanos,, A Novel Cysteine-rich Sequence-specific DNA-binding Protein Interacts with the Conserved X-box Motif of the Human Major Histocompatibility Complex Class II Genes Via a Repeated Cys-His Domain and Functions as a Transcriptional Repressor. 1994,180, 1763-1774.
  5. Yang, Y.; Li, H. G.; Wang, J.; Wang, H. L.; He, F.; Su, Y.; Zhang, Y.; Feng, C. H.; Niu, M.; Li, Z.; Liu, C.; Yin, W.; Xia, X., PeABF3 Enhances Drought Tolerance via Promoting ABA-induced Stomatal Closure by directly Regulating PeADF5 in Poplar. J Exp Bot 2020.
  6. Wang, D.; Meng, S.; Su, W.; Bao, Y.; Lu, Y.; Yin, W.; Liu, C.; Xia, X., Genome-Wide Analysis of Multiple Organellar RNA Editing Factor Family in Poplar Reveals Evolution and Roles in Drought Stress. Int J Mol Sci 2019,20, (6).
  7. Tuskan, G. A.; Difazio, S.; Jansson, S.; Bohlmann, J.; Grigoriev, I.; Hellsten, U.; Putnam, N.; Ralph, S.; Rombauts, S.; Salamov, A.; Schein, J.; Sterck, L.; Aerts, A.; Bhalerao, R. R.; Bhalerao, R. P.; Blaudez, D.; Boerjan, W.; Brun, A.; Brunner, A.; Busov, V.; Campbell, M.; Carlson, J.; Chalot, M.; Chapman, J.; Chen, G. L.; Cooper, D.; Coutinho, P. M.; Couturier, J.; Covert, S.; Cronk, Q.; Cunningham, R.; Davis, J.; Degroeve, S.; Dejardin, A.; Depamphilis, C.; Detter, J.; Dirks, B.; Dubchak, I.; Duplessis, S.; Ehlting, J.; Ellis, B.; Gendler, K.; Goodstein, D.; Gribskov, M.; Grimwood, J.; Groover, A.; Gunter, L.; Hamberger, B.; Heinze, B.; Helariutta, Y.; Henrissat, B.; Holligan, D.; Holt, R.; Huang, W.; Islam-Faridi, N.; Jones, S.; Jones-Rhoades, M.; Jorgensen, R.; Joshi, C.; Kangasjarvi, J.; Karlsson, J.; Kelleher, C.; Kirkpatrick, R.; Kirst, M.; Kohler, A.; Kalluri, U.; Larimer, F.; Leebens-Mack, J.; Leple, J. C.; Locascio, P.; Lou, Y.; Lucas, S.; Martin, F.; Montanini, B.; Napoli, C.; Nelson, D. R.; Nelson, C.; Nieminen, K.; Nilsson, O.; Pereda, V.; Peter, G.; Philippe, R.; Pilate, G.; Poliakov, A.; Razumovskaya, J.; Richardson, P.; Rinaldi, C.; Ritland, K.; Rouze, P.; Ryaboy, D.; Schmutz, J.; Schrader, J.; Segerman, B.; Shin, H.; Siddiqui, A.; Sterky, F.; Terry, A.; Tsai, C. J.; Uberbacher, E.; Unneberg, P.; Vahala, J.; Wall, K.; Wessler, S.; Yang, G.; Yin, T.; Douglas, C.; Marra, M.; Sandberg, G.; Van de Peer, Y.; Rokhsar, D., The genome of black cottonwood, Populus trichocarpa (Torr. & Gray). Science 2006,313, (5793), 1596-604.

Reviewer 3 Report

The authors identified four NFX1 genes from Populus trichocarpa, and investigated structural comparison including phylogeny, chromosomal localization and cis-acting elements between their protein sequences and from other plant species, such as Arabidopsis and rice. Furthermore, transcript level analysis of each organ and on response to phytohormones are carried out by qRT-PCR.

The results from experimental data that the authors provided in this Ms are clear and trustworthy, however, to publish in IJMS adaptive revision is necessary by the reason described below.

  1. ALL Figures and Tables are duplicated. Delete one in each figure and table.

  1. L55  Gibberellic acid is gibberellic acid.

  1. L58  What is the “X-box motif”? Add explanation.

  1. Figure 7  What is the reason of transcript level down of PtrNFX1-1 & 1-2 at 3 h after treatment of ABA?  Same expression pattern is shown in MeJA treated PtrNFX1-3.

  1. Which is correct, NF-X1 or NFX1?

  1. L440 What kind of plant line did the authors use in this study?

Author Response

manuscript ID (ijms-1201645)

Dear Editor:

We would like to thank you and the reviewers for your valuable comments and suggestions that helped us further our understanding of some aspects on this field and to improve our manuscript. After carefully considering the Reviewers’ comments, we revised our manuscript by point-by-point, and hereby list our responses to these comments. On the basis of the previous version, we provided a trace version and re-uploaded it.

Please do not hesitate to contact me in case the manuscript has any further issues.

Thank you very much for your help.

We are looking forward to hearing from you soon.

Yours sincerely,

Fang He, Ph.D.

Associate professor of forest tree genetics and breeding

College of Forestry,

Sichuan Agricultural University , Chengdou 611130 , China

Tel: +86-17683777884

E-mail: [email protected][email protected]

Responses to reviewer 3:

【Comment 1】ALL Figures and Tables are duplicated. Delete one in each figure and table.  

[Our response 1]: Thank you for your advice.

In accordance with the requirements of the journal, the PDF version of the previous upload has retained the modification traces, so many diagrams and tables appear twice. We have deleted the redundant figure and table, and the revised manuscript this time only retains the traces of this revision. Thanks again for your advice.

【Comment 2】L55  Gibberellic acid is gibberellic acid.

[Our response 2]: Thank you for your advice. We have rewritten the word.

As nicely advised, the introduction have been updated in this paper.

The original is “To be specific, as signal molecules, … plant responses to stress”, and we have revised it to “To be specific, as signal molecules, hormones such as abscisic acid (ABA), ethylene (ET) and gibberellic acid (GA) can mediate signaling pathways, which are mostly relied by TFs regulating plant responses to stress” (in line 48-51).

【Comment 3】L58  What is the “X-box motif”? Add explanation.

[Our response 3]: Thank you for your question. We rewrote the sentence to make the purpose of the article clearer. "X-box motif" is the functional element on the promoter of Class II MHC (major histocompatibility Complex) gene [4]. We rewrote the sentence to make it clearer.

As nicely advised, the introduction have been updated in this paper.

 The original is “NF-X1 was first d… and inhibits its expression”, and we have revised it to “NF-X1 was first discovered in 1994 and has been identified as a protein that binds to the promoter (X-Box motif) of the CLASS II MHC (major histocompatibility Complex) gene and inhibits its expression” (in line 54-56).

【Comment 4】Figure 7  What is the reason of transcript level down of PtrNFX1-1 & 1-2 at 3 h after treatment of ABA?  Same expression pattern is shown in MeJA treated PtrNFX1-3. 

[Our response 4]: Thank you for your question. The reviewer found that PtrNFX1-1/2 had a downward trend at 3 hours after treatment of ABA. But compared with normal, PtrNFX1-1/2 was activated after 3 hours. Firstly, genes cannot be expressed at high levels all the time. In the case of the same external factors, it may also be affected by internal plant factors, including the regulation of other genes and the modification of proteins. In addition, similar results have appeared in many papers. For example, under the treatment of mannitol, the expression of PeABF3 gene in poplars was up-regulated at the third hour, down-regulated at the sixth hour, and up-regulated again at the ninth hour[5]. Similarly, in the International Journal of Molecular Sciences, the transcription level of PtrMORF8.1 in poplar trees under drought treatment was upregulated at 3 h, decreased at 6 h, and upregulated again at 9 h[6].

【Comment 5】Which is correct, NF-X1 or NFX1? 

[Our response 5]: Thank you for your question. NF-X1 is correct.

The original is “NFX1”, and we have revised it to “NF-X1” (in line 77).

The original is “NFX1”, and we have revised it to “NF-X1” (in line 83).

The original is “NFX1”, and we have revised it to “NF-X1” (in line 336).

The original is “NFX1”, and we have revised it to “NF-X1” (in line 337).

The original is “TaNFX1-3”, and we have revised it to “TaNF-X1-3” (in line 72).

The original is “TaNFX1-2”, and we have revised it to “TaNF-X1-2” (in line 72).

The original is “AtNFXL1”, and we have revised it to “AtNF-XL1” (in line 64).

The original is “AtNFXL2”, and we have revised it to “AtNF-XL2” (in line 65).

The original is “AtNFXL1”, and we have revised it to “AtNF-XL1” (in line 67).

The original is “NFXL1”, and we have revised it to “NF-XL1” (in line 337).

【Comment 6】L440 What kind of plant line did the authors use in this study?

[Our response 6]: Thank you for your question. The plant material studied in this paper is Populus trichocarpa, whose whole genome was sequenced in 2006[7]. Because of its clear and complete genome, it is used as a woody model plant to study the gene function of poplar.

As nicely advised, the Materials and Methods have been updated in this paper.

“Because of its clear and complete genome, Populus trichocarpa is used as a woody model plant to study the gene function of poplar” (in line 416-417).

Thank you again for your questions and comments, which have been revised in the original text.

  1. He, F.; Niu, M. X.; Feng, C. H.; Li, H. G.; Su, Y.; Su, W. L.; Pang, H.; Yang, Y.; Yu, X.; Wang, H. L.; Wang, J.; Liu, C.; Yin, W.; Xia, X., PeSTZ1 confers salt stress tolerance by scavenging the accumulation of ROS through regulating the expression of PeZAT12 and PeAPX2 in Populus. Tree Physiol 2020.
  2. Zhou, J. M.; Zhang, Y., Plant Immunity: Danger Perception and Signaling. Cell 2020,181, (5), 978-989.
  3. Zhao, C.; Zhang, H.; Song, C.; Zhu, J.-K.; Shabala, S., Mechanisms of Plant Responses and Adaptation to Soil Salinity. The Innovation 2020,1, (1), 100017.
  4. Zhimin Song, S. K., $ Dimitris Thanos,, A Novel Cysteine-rich Sequence-specific DNA-binding Protein Interacts with the Conserved X-box Motif of the Human Major Histocompatibility Complex Class II Genes Via a Repeated Cys-His Domain and Functions as a Transcriptional Repressor. 1994,180, 1763-1774.
  5. Yang, Y.; Li, H. G.; Wang, J.; Wang, H. L.; He, F.; Su, Y.; Zhang, Y.; Feng, C. H.; Niu, M.; Li, Z.; Liu, C.; Yin, W.; Xia, X., PeABF3 Enhances Drought Tolerance via Promoting ABA-induced Stomatal Closure by directly Regulating PeADF5 in Poplar. J Exp Bot 2020.
  6. Wang, D.; Meng, S.; Su, W.; Bao, Y.; Lu, Y.; Yin, W.; Liu, C.; Xia, X., Genome-Wide Analysis of Multiple Organellar RNA Editing Factor Family in Poplar Reveals Evolution and Roles in Drought Stress. Int J Mol Sci 2019,20, (6).
  7. Tuskan, G. A.; Difazio, S.; Jansson, S.; Bohlmann, J.; Grigoriev, I.; Hellsten, U.; Putnam, N.; Ralph, S.; Rombauts, S.; Salamov, A.; Schein, J.; Sterck, L.; Aerts, A.; Bhalerao, R. R.; Bhalerao, R. P.; Blaudez, D.; Boerjan, W.; Brun, A.; Brunner, A.; Busov, V.; Campbell, M.; Carlson, J.; Chalot, M.; Chapman, J.; Chen, G. L.; Cooper, D.; Coutinho, P. M.; Couturier, J.; Covert, S.; Cronk, Q.; Cunningham, R.; Davis, J.; Degroeve, S.; Dejardin, A.; Depamphilis, C.; Detter, J.; Dirks, B.; Dubchak, I.; Duplessis, S.; Ehlting, J.; Ellis, B.; Gendler, K.; Goodstein, D.; Gribskov, M.; Grimwood, J.; Groover, A.; Gunter, L.; Hamberger, B.; Heinze, B.; Helariutta, Y.; Henrissat, B.; Holligan, D.; Holt, R.; Huang, W.; Islam-Faridi, N.; Jones, S.; Jones-Rhoades, M.; Jorgensen, R.; Joshi, C.; Kangasjarvi, J.; Karlsson, J.; Kelleher, C.; Kirkpatrick, R.; Kirst, M.; Kohler, A.; Kalluri, U.; Larimer, F.; Leebens-Mack, J.; Leple, J. C.; Locascio, P.; Lou, Y.; Lucas, S.; Martin, F.; Montanini, B.; Napoli, C.; Nelson, D. R.; Nelson, C.; Nieminen, K.; Nilsson, O.; Pereda, V.; Peter, G.; Philippe, R.; Pilate, G.; Poliakov, A.; Razumovskaya, J.; Richardson, P.; Rinaldi, C.; Ritland, K.; Rouze, P.; Ryaboy, D.; Schmutz, J.; Schrader, J.; Segerman, B.; Shin, H.; Siddiqui, A.; Sterky, F.; Terry, A.; Tsai, C. J.; Uberbacher, E.; Unneberg, P.; Vahala, J.; Wall, K.; Wessler, S.; Yang, G.; Yin, T.; Douglas, C.; Marra, M.; Sandberg, G.; Van de Peer, Y.; Rokhsar, D., The genome of black cottonwood, Populus trichocarpa (Torr. & Gray). Science 2006,313, (5793), 1596-604.

This manuscript is a resubmission of an earlier submission. The following is a list of the peer review reports and author responses from that submission.

Round 1

Reviewer 1 Report

Dear colleagues, your manuscript seems to be an original, high quality research work. However, I suggest to revise it moderately because of some formal and structural causes.

First of all, please define your goals clearly at the end of the 'Introduction' section. In the present form, readers of the text don't know what exactly you want to know  with this study.

Moreover, please revise ''Materials and Methods' section, especially the 4.1. and 4.7. parts. In 4.1 the genome sequences used in this work should be referred exactly. In Phytozone database are at least two A. thaliana and two O. sativa genomes. Which ones did you use? In 4.7 please clearly define the treatments. How ABA, PEG, JA were applied? What does 'normal circumstances' mean? Same seedling plants without treatments? How could you separate young, mature and old leaves in a 60 days old seedling?

Please use the scientific name of plants more consistently: Glycine max not Glycine Max (e.g. row 116) , etc. Furthermore, don't use Populus instead of P. trichocarpa (in the rows 75-76 - the widely used model organism is P. trichocarpa, not the Populus genus). In some cases, I suggest using poplar instead of populus (e.g. in the row 164), because writing 'populus' is not correct. Please write the full scientific name of a given organism at the first time: there is not any reference, that 'O. sativa' means Oryza sativa in the text.

Finally, I suggest asking an English reviewer to correct the text, because there are many incorrect phrases in the text (e.g. row 16: 'was performed' instead of 'was finished', row 44: 'activate' instead of 'active', etc.).

Author Response

We would like to thank you and the reviewers for your valuable comments and suggestions that helped us further our understanding of some aspects on this field and to improve our manuscript. After carefully considering the Reviewers’ comments, we revised our manuscript by point-by-point, and hereby list our responses to these comments.

Due to the journal word limit, some methodological steps were simplified in our first manuscript. To address your concerns, we have altered the manuscript to include more background information and clearly indicate our objective, more detailed methods and materials, and more detailed descriptions of the experimental results, more adequate data interpretation. The manuscript was sent to a professional English reviewer for correct the text.

Please do not hesitate to contact me in case the manuscript has any further issues.

Thank you very much for your help.

We are looking forward to hearing from you soon.

Yours sincerely,

Fang He, Ph.D.

Associate professor of forest tree genetics and breeding

College of Forestry,

Sichuan Agricultural University , Chengdou 611130 , China

Tel: +86-17683777884

E-mail: [email protected][email protected]

-------------------------------------------------------------------------------------------------------

Responses to reviewer 1:

【Comment 1】First of all, please define your goals clearly at the end of the 'Introduction' section. In the present form, readers of the text don't know what exactly you want to know with this study.

[Our response1]: Thank you for your advice. We have added the goal at the end of the 'Introduction' section to let the reader know the meaning of our work.

Presently, most of the studies on NFX1 transcription factors are focused on human and bacteria. There are few functional studies on NF-X1 transcription factors for biological and abiotic stress in poplar trees. Four putative NF-X1 genes in P. trichocarpa were studied from the aspects of molecular structure, phylogeny, chromosomal localization, and expression pattern (under ABA, PEG, and JA treatment), which provided genetic resources and theoretical basis for forest genetic engineering and was of great significance to cultivate poplar with high resistance to adversity.

As nicely advised, the 'Introduction' section update in this paper.

The original is “There are few functional studies on NF-X1…development and stress.”, and we have revised it to “Presently, most of the studies on NFX1 transcription factors are focused on human and bacteria. There are few functional studies on NF-X1 transcription factors for biological and abiotic stress in poplar trees. Four putative NF-X1 genes in P. trichocarpa were studied from the aspects of molecular structure, phylogeny, chromosomal localization, and expression pattern (under ABA, PEG, and JA treatment), which provided genetic resources and theoretical basis for forest genetic engineering and was of great significance to cultivate poplar with high resistance to adversity.” (in line 82-88).

【Comment 2】Moreover, please revise ''Materials and Methods' section, especially the 4.1. and 4.7. parts. In 4.1 the genome sequences used in this work should be referred exactly. In Phytozone database are at least two A. thaliana and two O. sativa genomes. Which ones did you use?

[Our response 2]: Thank you for your advice and question. Athaliana_447_TAIR10.fa.gz and Osativa_323_v7.0.fa.gz were used to A. thaliana and two O. sativa genomes, respectively. The two genomes are now well spliced and complete, and are widely used in plant comparative genomics and molecular biology studies. In order to describe the methods and materials in more detail, we have rewritten two section.

As nicely advised, the methods and materials update in this paper.

The original is “The genome data of A.thaliana …Phytozome database”, and we have revised it to “The genome data of A.thaliana (Athaliana_447_TAIR10.fa.gz), O.sativa (Osativa_323_v7.0.fa.gz), and P. trichocarpa (Ptrichocarpa_533_v4.0.fa.gz) were obtained from Phytozome database (https://genome.jgi.doe.gov/portal/pages).” (in line 351-353).

【Comment 3】In 4.7 please clearly define the treatments. How ABA, PEG, JA were applied?

[Our response 3]: Thank you for your advice. We added the the ABA, PEG, JA treatment method as required. In order to describe the treatments in more detail, we have rewritten this paragraph.

The 60-day-old seedlings were treated with PEG, ABA, and JA treatment. There are at least 5 biological repeats in the gradient of each treatment. For each treatment, please refer to previously published papers with appropriate revisions [1, 2]. For drought treatments, similarly grown seedlings of P.trichocarpa (40–50 cm high, with 30–35 leaves)were subjected to a 15% PEG6000 solution for 0, 1, 3, 6, 9 and 12 h. For phytohormone analysis, similarly grown seedlings of P.trichocarpa were treated with a solution containing 100 μM ABA (Sigma, A1049) and MeJA (Sigma, 678406)  , respectively, for 0, 1, 3, 6, 9 and 12 h. At the time point of various treatments, the leaves were separated from the plant, rapidly freezed in liquid nitrogen and deposited in ultra low temperature freezer.

As nicely advised, the methods and materials update in this paper.

The original is “The 60-day-old seedlings were…temperature freezer.”, and we have revised it to “The 60-day-old seedlings were treated with PEG, ABA, and JA treatment. There are at least 5 biological repeats in the gradient of each treatment. For each treatment, please refer to previously published papers with appropriate revisions [1, 2]. For PEG treatments, similarly grown seedlings of P.trichocarpa (40–50 cm high, with 30–35 leaves)were subjected to a 15% PEG6000 solution for 0, 1, 3, 6, 9 and 12 h. For phytohormone analysis, similarly grown seedlings of P.trichocarpa were treated with a solution containing 100 μM ABA (Sigma, A1049) and MeJA (Sigma, 678406) , respectively, for 0, 1, 3, 6, 9 and 12 h. At the time point of various treatments, the leaves were separated from the plant, rapidly freezed in liquid nitrogen and deposited in ultra low temperature freezer.” (in line 410-418).

【Comment 4】What does 'normal circumstances' mean? Same seedling plants without treatments?

[Our response 4]: 

As nicely advised, the methods and materials update in this paper.

The original is “Under normal circumstances…cryogenic icebox.”, and we have revised it to “In the absence of any stress, plant tissues and organs were sampled and each plant tissue and organ had at least 5 biological replicates. Finally, we collected different organs and tissues of 2-month old P.trichocarpa, including young leaves, mature leaves, old leaves, xylem, phloem, and roots at the same time and immediately immersed them in liquid nitrogen.” (in line 420-424).

【Comment 5】How could you separate young, mature and old leaves in a 60 days old seedling?

[Our response 5]: Thank you for your question. In order to describe the sampling in more detail, we have rewritten this sentence.

we defined the 1 to 12 leaves(from top) as young leaves, 13 to 30 leaves as mature leaves ( the highest net photosynthetic rate), and 31-40 leaves as old leaves according to the net photosynthetic rate of 2 month old poplar.

The original is “Under normal circumstances…cryogenic icebox.”, and we have revised it to “For the sampling of poplar tissues and organs, please refer to a previously published paper[1]. In the absence of any stress, plant tissues and organs were sampled and each plant tissue and organ had at least 5 biological replicates. Finally, we collected different organs and tissues of 2-month old P.trichocarpa, including young leaves, mature leaves, old leaves, xylem, phloem, and roots at the same time and immediately immersed them in liquid nitrogen” (in line 419-424)

【Comment 6】Please use the scientific name of plants more consistently: Glycine max not Glycine Max (e.g. row 116) , etc.

[Our response 6]: Thank you for your advice. We have checked the Latin names of all the species in the paper and standardized spelling.

The original is “Glycine Max ”, and we have revised it to “Glycine max(in line 122) .

The original is “O.sativa”, and we have revised it to “Oryza sativa(in line 90) .

The original is “Arabidopsis thaliana”, and we have revised it to “A.thaliana(in line 296) .

The original is “rice”, and we have revised it to “O. sativa(in line 296) .

The original is “Glycine Max ”, and we have revised it to “Glycine max(in line 298) .

【Comment 7】Furthermore, don't use Populus instead of P. trichocarpa (in the rows 75-76 - the widely used model organism is P. trichocarpa, not the Populus genus). In some cases, I suggest using poplar instead of populus (e.g. in the row 164), because writing 'populus' is not correct.

[Our response 7]: Thank you for your advice. We have corrected the names of all Populus genus in the text.

The original is “populus”, and we have revised it to “poplar(in line 11) .

The original is “populus”, and we have revised it to “poplar(in line 16) .

The original is “populus”, and we have revised it to “Populus trichocarpa(in line 29) .

The original is “populus”, and we have revised it to “P. trichocarpa(in line 78) .

The original is “populus”, and we have revised it to “P. trichocarpa(in line 79) .

The original is “populus”, and we have revised it to “poplar(in line 160) .

The original is “populus”, and we have revised it to “P. trichocarpa(in line 172) .

The original is “populus”, and we have revised it to “P. trichocarpa(in line 173) .

The original is “populus”, and we have revised it to “P. trichocarpa(in line 178) .

The original is “populus”, and we have revised it to “P. trichocarpa(in line 179) .

The original is “populus”, and we have revised it to “P. trichocarpa(in line 180) .

The original is “populus”, and we have revised it to “poplar(in line 185) .

The original is “populus”, and we have revised it to “poplar(in line 186) .

The original is “populus”, and we have revised it to “poplar(in line 211) .

The original is “populus”, and we have revised it to “poplar(in line 215) .

The original is “populus”, and we have revised it to “poplar(in line 249) .

The original is “populus”, and we have revised it to “poplar(in line 277) .

The original is “populus”, and we have revised it to “poplar(in line 307) .

The original is “populus”, and we have revised it to “poplar(in line 311) .

The original is “populus”, and we have revised it to “poplar(in line 381) .

The original is “populus”, and we have revised it to “poplar(in line 441) .

【Comment 8】Please write the full scientific name of a given organism at the first time: there is not any reference, that 'O. sativa' means Oryza sativa in the text.

[Our response 8]: As nicely advised, we have updated the full name of the acronym that appears for the first time in this article. We have listed the exact location of the acronym update.

The original is “O. sativa”, and we have revised it to “Oryza sativa(in line 90) .

【Comment 9】Finally, I suggest asking an English reviewer to correct the text, because there are many incorrect phrases in the text (e.g. row 16: 'was performed' instead of 'was finished', row 44: 'activate' instead of 'active', etc.). 

[Our response 9]: Thank you for your comments on the original manuscript. We have corrected the grammatical mistakes and the manuscript was sent to an English reviewer to correct the text. 

The original is “In the recent research…family was finished”, and we have revised it to “In this study, whole genome analysis of the poplar NF-X1 family was performed and 4 PtrNF-X1 genes were identified.(in line 17-18) .

The original is “Furthermore, TFs can … signal cascade”, and we have revised it to “Furthermore, TFs can interact with cis-acting elements in the promoter region of stress response genes to activate the entire gene network of signal cascade”(in line 44-46) .

Thank you for your comments. All questions are answered in the text.

References

  1. He, F.; Li, H. G.; Wang, J. J.; Su, Y.; Wang, H. L.; Feng, C. H.; Yang, Y.; Niu, M. X.; Liu, C.; Yin, W.; Xia, X., PeSTZ1, a C2H2-type zinc finger transcription factor from Populus euphratica, enhances freezing tolerance through modulation of ROS scavenging by directly regulating PeAPX2. Plant Biotechnol J 2019.
  2. Xie, T.; Chen, C.; Li, C.; Liu, J.; Liu, C.; He, Y., Genome-wide investigation of WRKY gene family in pineapple: evolution and expression profiles during development and stress. BMC Genomics 2018,19, (1), 490.
  3. Wang, D.; Meng, S.; Su, W.; Bao, Y.; Lu, Y.; Yin, W.; Liu, C.; Xia, X., Genome-Wide Analysis of Multiple Organellar RNA Editing Factor Family in Poplar Reveals Evolution and Roles in Drought Stress. Int J Mol Sci 2019,20, (6).
  4. Li, M.; Wang, R.; Liu, Z.; Wu, X.; Wang, J., Genome-wide identification and analysis of the WUSCHEL-related homeobox (WOX) gene family in allotetraploid Brassica napus reveals changes in WOX genes during polyploidization. BMC Genomics 2019,20, (1), 317.
  5. Singh, V. K.; Rajkumar, M. S.; Garg, R.; Jain, M., Genome-wide identification and co-expression network analysis provide insights into the roles of auxin response factor gene family in chickpea. Sci Rep 2017,7, (1), 10895.
  6. Zhu, J. K., Abiotic Stress Signaling and Responses in Plants. Cell 2016,167, (2), 313-324.
  7. Hewage, K. A. H.; Yang, J. F.; Wang, D.; Hao, G. F.; Yang, G. F.; Zhu, J. K., Chemical Manipulation of Abscisic Acid Signaling: A New Approach to Abiotic and Biotic Stress Management in Agriculture. Adv Sci (Weinh) 2020,7, (18), 2001265.
  8. Zhao, D.; Wang, H.; Chen, S.; Yu, D.; Reiter, R. J., Phytomelatonin: An Emerging Regulator of Plant Biotic Stress Resistance. Trends Plant Sci 2020.
  9. Takahashi, F.; Shinozaki, K., Long-distance signaling in plant stress response. Curr Opin Plant Biol 2019,47, 106-111.
  10. Lozano-Juste, J.; Alrefaei, A. F.; Rodriguez, P. L., Plant Osmotic Stress Signaling: MAPKKKs Meet SnRK2s. Trends Plant Sci 2020.
  11. Lin, Z.; Li, Y.; Zhang, Z.; Liu, X.; Hsu, C. C.; Du, Y.; Sang, T.; Zhu, C.; Wang, Y.; Satheesh, V.; Pratibha, P.; Zhao, Y.; Song, C. P.; Tao, W. A.; Zhu, J. K.; Wang, P., A RAF-SnRK2 kinase cascade mediates early osmotic stress signaling in higher plants. Nat Commun 2020,11, (1), 613.
  12. Li, J.; Li, Y.; Yin, Z.; Jiang, J.; Zhang, M.; Guo, X.; Ye, Z.; Zhao, Y.; Xiong, H.; Zhang, Z.; Shao, Y.; Jiang, C.; Zhang, H.; An, G.; Paek, N. C.; Ali, J.; Li, Z., OsASR5 enhances drought tolerance through a stomatal closure pathway associated with ABA and H2 O2 signalling in rice. Plant Biotechnol J 2017,15, (2), 183-196.
  13. Liu, J.; Zhang, C.; Wei, C.; Liu, X.; Wang, M.; Yu, F.; Xie, Q.; Tu, J., The RING Finger Ubiquitin E3 Ligase OsHTAS Enhances Heat Tolerance by Promoting H2O2-Induced Stomatal Closure in Rice. Plant Physiol 2016,170, (1), 429-43.
  14. He, F.; Wang, H. L.; Li, H. G.; Su, Y.; Li, S.; Yang, Y.; Feng, C. H.; Yin, W.; Xia, X., PeCHYR1, a ubiquitin E3 ligase from Populus euphratica, enhances drought tolerance via ABA-induced stomatal closure by ROS production in Populus. Plant Biotechnol J 2018.
  15. Lim, C. W.; Baek, W.; Lee, S. C., The Pepper RING-Type E3 Ligase CaAIRF1 Regulates ABA and Drought Signaling via CaADIP1 Protein Phosphatase Degradation. Plant Physiol 2017,173, (4), 2323-2339.
  16. Duan, M.; Zhang, R.; Zhu, F.; Zhang, Z.; Gou, L.; Wen, J.; Dong, J.; Wang, T., A Lipid-Anchored NAC Transcription Factor Is Translocated into the Nucleus and Activates Glyoxalase I Expression during Drought Stress. Plant Cell 2017,29, (7), 1748-1772.
  17. Mussig, C.; Schroder, F.; Usadel, B.; Lisso, J., Structure and putative function of NFX1-like proteins in plants. Plant Biol (Stuttg) 2010,12, (3), 381-94.
  18. He, F.; Niu, M. X.; Feng, C. H.; Li, H. G.; Su, Y.; Su, W. L.; Pang, H.; Yang, Y.; Yu, X.; Wang, H. L.; Wang, J.; Liu, C.; Yin, W.; Xia, X., PeSTZ1 confers salt stress tolerance by scavenging the accumulation of ROS through regulating the expression of PeZAT12 and PeAPX2 in Populus. Tree Physiol 2020.

Reviewer 2 Report

Authors researched identification of four NF-X1 genes in Populus trichocarpa (PtrNF-X1) by their gene structure, phylogeny, chromosomal localization, gene replication with Cis-elements, and expression patterns. Authors suggested that the evolution mechanism from two PtrNF-X1 genes (such as PtrNF-X1-1 and PtrNF-X1-2) may be related with those synteny blocks of Arabidopsis (ATNF-X1-2) and rice (OsNF-X1-1) by the gene replication. They are also classified in class I with evolutionary and phylogenic analysis of the 138 NF-X1 proteins in the 52 plant species. The five NF-X1 proteins from Salix purpurea were adjacent to the PtrNF-X1 genes. Are compared with the NF-X1 genes from gene structure, conserved and motifs? Why is compared with between the eight NF-X1 genes from Arabidopsis, rice, and Populus trichocarpa?

How is explained the relationship between the expression levels of the four NF-X1 genes in root drought, beetle damage, and mechanical damage and responses to PEG and ABA ? Why is useful to select the candidate NF-X1 genes for stress resistance to PEG and ABA?

Author Response

We would like to thank you and the reviewers for your valuable comments and suggestions that helped us further our understanding of some aspects on this field and to improve our manuscript. After carefully considering the Reviewers’ comments, we revised our manuscript by point-by-point, and hereby list our responses to these comments.

Due to the journal word limit, some methodological steps were simplified in our first manuscript. To address your concerns, we have altered the manuscript to include more background information and clearly indicate our objective, more detailed methods and materials, and more detailed descriptions of the experimental results, more adequate data interpretation. The manuscript was sent to a professional English reviewer for correct the text.

Please do not hesitate to contact me in case the manuscript has any further issues.

Thank you very much for your help.

We are looking forward to hearing from you soon.

Yours sincerely,

Fang He, Ph.D.

Associate professor of forest tree genetics and breeding

College of Forestry,

Sichuan Agricultural University , Chengdou 611130 , China

Tel: +86-17683777884

E-mail: [email protected][email protected]

-------------------------------------------------------------------------------------------------------

Responses to reviewer 2:

【Comment 1】The five NF-X1 proteins from Salix purpurea were adjacent to the PtrNF-X1 genes. Are compared with the NF-X1 genes from gene structure, conserved and motifs?

As the reviewer noted, the five NF-X1 proteins from Salix purpurea were adjacent to the PtrNF-X1 proteins. First of all, poplars and willows belong to the same family of plant (Salicaceae Mirb.). Moreover, evolutionary trees are constructed according to amino acid sequence alignment, so their protein conserved sequence and conserved motif are the most similar. Therefore, it is not of great significance to compare their gene structure, conserved motif and conserved motif with willow. So we only compared model plants Arabidopsis thaliana, rice, and poplar.

[Our response1]: Thank you for your question. We have discussed it in text. As the reviewer noted, the five NF-X1 proteins from Salix purpurea were adjacent to the PtrNF-X1 proteins. First of all, poplars and willows belong to the same family of plant (Salicaceae Mirb.). Moreover, evolutionary trees are constructed according to amino acid sequence alignment (The amino acids of all NF-X1 target sequences were analyzed by ClustalX), so their protein conserved structure and conserved motif are the most similar. Therefore, it is not of great significance to compare their gene structure, conserved motif and conserved motif with Salix purpurea. Finally, we selected model plants (plants Arabidopsis thaliana, rice, and poplar) for comparison of gene structure, conserved protein domain, and conserved motif.

The original is “Referencing previous gene … chromosomal locations (Table 1 and Figure 4a).”, and we have revised it to “Referencing previous gene family studies [3], the NF-X1 genes in poplars were named as PtrNF-X1-1 to PtrNF-X1-4 according to their chromosomal locations (Table 1 and Figure 4a).The model plants Arabidopsis thaliana, rice, and poplar were applied to the comparative genomics of NF-X1 family” (in line 275-278)

The original is “Because poplar … same department ”, and we have revised it to “Because poplar and willow belong to the same department [3, 4], the five NF-X1 proteins from Salix purpurea were adjacent to the PtrNF-X1 proteins.” (in line294-296)

【Comment 2】Why is compared with between the eight NF-X1 genes from Arabidopsis, rice, and Populus trichocarpa?

[Our response 2]: Thank you for your question. We have discussed it in the paper.

We have compared the NFX1 gene from the aspects of gene structure, protein conserved domain and conserved sequence. Firstly, rice is a monocotyledonous model plant, Arabidopsis is a dicotyledonous model plant, and Populus trichocarpa is a perennial woody model plant. All three of them had well-patched and complete genomic data. Second, these three species have been used for comparative Genomics studies in many excellent journals[2, 3], such as International Journal of Molecular Sciences, and BMC Genomics. Thirdly, these three species have been used simultaneously in many articles for comparative genomics. Genome-Wide Analysis of Multiple Organellar RNA Editing Factor Family in rice, Arabidopsis, and Populus trichocarpa [3]. At same time, these three species have also been included in genome-wide studies of the WRKY gene family [5].

The method and introduction have been updated this paper.

“The genomes of model plants (such as rice, Arabidopsis, poplar, etc.) can reveal the function of plant genes [1, 2]. Meanwhile, comparative genomics is used to study the NFX1 gene and compare its genome structure to understand its function and expression mechanism.” (in line74-77).

“As these three model plants can all be searched for well-patched and complete genomic data[2, 3], rice, Arabidopsis, and Populus trichocarpa will be selected for the study of gene structure, protien structure and motifs.” (in line377-379).

【Comment 3】How is explained the relationship between the expression levels of the four NF-X1 genes in root drought, beetle damage, and mechanical damage and responses to PEG and ABA ?

 [Our response 3]: Thank you for your question. We have discussed it in text.

Firstly, we analyzed transcriptome responses to drought in poplar roots and leaves, beetle damage in poplar leaves, and mechanical damage in poplar leaves. Many excellent reviews have shown that drought and mechanical damage in plants are the most common abiotic stresses[6, 7], and beetle damage are the most important biological stresses[8, 9]. In addition, osmotic stress is the main signal of abiotic stress in plants[10, 11], including drought, salt, cold and heat, etc. Furthermore, previously published papers have shown that PEG can mimic osmotic stress in plants[2, 12, 13].

Secondly, many excellent reviews have indicated that ABA signaling pathway is the core pathway for regulating plant response to stress[6, 7]. Similarly, ABA signaling is involved in stress responses to drought stress and mechanical damage in poplars. In summary, the expression level of NFX1 under PEG and ABA treatment was selected to reflect the transcription level of NFX1 in poplar drought stress, beetle damage, and mechanical damage.

The introduction, method, and discussion have been updated this paper.

“In addition, osmotic stress is the main signal of abiotic stress in plants[10, 11], including drought, salt, cold and heat, etc.” (in line39-40)

“Moreover , many excellent reviews have indicated that ABA signaling pathway is the core pathway for regulating plant response to stress [6, 7]. Similarly, ABA signaling is involved in stress responses to drought stress, beetle damage, and mechanical damage in poplars (Figure 7c). In summary, the expression level of NFX1 under PEG and ABA treatment was selected to reflect the transcription level of NFX1 in poplar drought stress, beetle damage, and mechanical damage.” (in line327-333)

“The previously published papers have shown that PEG and ABA treatments were used to investigate gene responses to abiotic stresses in plants [12, 14-16].” (in line408-409)

【Comment 4】Why is useful to select the candidate NF-X1 genes for stress resistance to PEG and ABA? 

[Our response 4]: This is a very important question for us. We have discussed it in text. 

Presently, most of the studies on NFX1 are focused on human and bacteria [17]. There are few functional studies on NF-X1 transcription factors for biological and abiotic stress in poplar trees. Firstly, the abiotic stresses of plants include water shortage, salt stress and cold stress, etc [1, 14, 18]. At the same time, osmotic stress as the main signal of abiotic stress also affects the growth and development of this plant[10, 11]. Furthermore, previously published papers have shown that PEG can mimic osmotic stress in plants[2, 12, 13]. Second, abscisic acid is the core signaling pathway of plant stress, which is essential for the study of abiotic stress in plants[6, 7]. Third, many highly cited papers have used PEG and ABA treatments to investigate gene responses to abiotic stresses in plants[12, 14-16]. In conclusion, four NF-X1 genes in P. trichocarpa were studied from the expression pattern (under ABA and PEG treatment), which provided genetic resources and theoretical basis for forest genetic engineering and was of great significance to cultivate poplar with high resistance to adversity.

The introduction, method, and discussion have been updated this paper.

“In addition, osmotic stress is the main signal of abiotic stress in plants[10, 11], including drought, salt, cold and heat, etc. ” (in line39-40)

“Four putative NF-X1 genes in P. trichocarpa were studied from the aspects of molecular structure, phylogeny, chromosomal localization, and expression pattern (under ABA, PEG, and JA treatment), which provided genetic resources and theoretical basis for forest genetic engineering and was of great significance to cultivate poplar with high resistance to adversity.” (in line 84-88)

“Moreover , many excellent reviews have indicated that ABA signaling pathway is the core pathway for regulating plant response to stress [6, 7]. Similarly, ABA signaling is involved in stress responses to drought stress, beetle damage, and mechanical damage in poplars (Figure 7c). In summary, the expression level of NFX1 under PEG and ABA treatment was selected to reflect the transcription level of NFX1 in poplar drought stress, beetle damage, and mechanical damage.” (in line327-333)

“The previously published papers have shown that PEG and ABA treatments were used to investigate gene responses to abiotic stresses in plants [12, 14-16].” (in line408-409)

Thank you for your comments. All questions are answered in the text.

-------------------------------------------------------------------------------------------------------

References

  1. He, F.; Li, H. G.; Wang, J. J.; Su, Y.; Wang, H. L.; Feng, C. H.; Yang, Y.; Niu, M. X.; Liu, C.; Yin, W.; Xia, X., PeSTZ1, a C2H2-type zinc finger transcription factor from Populus euphratica, enhances freezing tolerance through modulation of ROS scavenging by directly regulating PeAPX2. Plant Biotechnol J 2019.
  2. Xie, T.; Chen, C.; Li, C.; Liu, J.; Liu, C.; He, Y., Genome-wide investigation of WRKY gene family in pineapple: evolution and expression profiles during development and stress. BMC Genomics 2018,19, (1), 490.
  3. Wang, D.; Meng, S.; Su, W.; Bao, Y.; Lu, Y.; Yin, W.; Liu, C.; Xia, X., Genome-Wide Analysis of Multiple Organellar RNA Editing Factor Family in Poplar Reveals Evolution and Roles in Drought Stress. Int J Mol Sci 2019,20, (6).
  4. Li, M.; Wang, R.; Liu, Z.; Wu, X.; Wang, J., Genome-wide identification and analysis of the WUSCHEL-related homeobox (WOX) gene family in allotetraploid Brassica napus reveals changes in WOX genes during polyploidization. BMC Genomics 2019,20, (1), 317.
  5. Singh, V. K.; Rajkumar, M. S.; Garg, R.; Jain, M., Genome-wide identification and co-expression network analysis provide insights into the roles of auxin response factor gene family in chickpea. Sci Rep 2017,7, (1), 10895.
  6. Zhu, J. K., Abiotic Stress Signaling and Responses in Plants. Cell 2016,167, (2), 313-324.
  7. Hewage, K. A. H.; Yang, J. F.; Wang, D.; Hao, G. F.; Yang, G. F.; Zhu, J. K., Chemical Manipulation of Abscisic Acid Signaling: A New Approach to Abiotic and Biotic Stress Management in Agriculture. Adv Sci (Weinh) 2020,7, (18), 2001265.
  8. Zhao, D.; Wang, H.; Chen, S.; Yu, D.; Reiter, R. J., Phytomelatonin: An Emerging Regulator of Plant Biotic Stress Resistance. Trends Plant Sci 2020.
  9. Takahashi, F.; Shinozaki, K., Long-distance signaling in plant stress response. Curr Opin Plant Biol 2019,47, 106-111.
  10. Lozano-Juste, J.; Alrefaei, A. F.; Rodriguez, P. L., Plant Osmotic Stress Signaling: MAPKKKs Meet SnRK2s. Trends Plant Sci 2020.
  11. Lin, Z.; Li, Y.; Zhang, Z.; Liu, X.; Hsu, C. C.; Du, Y.; Sang, T.; Zhu, C.; Wang, Y.; Satheesh, V.; Pratibha, P.; Zhao, Y.; Song, C. P.; Tao, W. A.; Zhu, J. K.; Wang, P., A RAF-SnRK2 kinase cascade mediates early osmotic stress signaling in higher plants. Nat Commun 2020,11, (1), 613.
  12. Li, J.; Li, Y.; Yin, Z.; Jiang, J.; Zhang, M.; Guo, X.; Ye, Z.; Zhao, Y.; Xiong, H.; Zhang, Z.; Shao, Y.; Jiang, C.; Zhang, H.; An, G.; Paek, N. C.; Ali, J.; Li, Z., OsASR5 enhances drought tolerance through a stomatal closure pathway associated with ABA and H2 O2 signalling in rice. Plant Biotechnol J 2017,15, (2), 183-196.
  13. Liu, J.; Zhang, C.; Wei, C.; Liu, X.; Wang, M.; Yu, F.; Xie, Q.; Tu, J., The RING Finger Ubiquitin E3 Ligase OsHTAS Enhances Heat Tolerance by Promoting H2O2-Induced Stomatal Closure in Rice. Plant Physiol 2016,170, (1), 429-43.
  14. He, F.; Wang, H. L.; Li, H. G.; Su, Y.; Li, S.; Yang, Y.; Feng, C. H.; Yin, W.; Xia, X., PeCHYR1, a ubiquitin E3 ligase from Populus euphratica, enhances drought tolerance via ABA-induced stomatal closure by ROS production in Populus. Plant Biotechnol J 2018.
  15. Lim, C. W.; Baek, W.; Lee, S. C., The Pepper RING-Type E3 Ligase CaAIRF1 Regulates ABA and Drought Signaling via CaADIP1 Protein Phosphatase Degradation. Plant Physiol 2017,173, (4), 2323-2339.
  16. Duan, M.; Zhang, R.; Zhu, F.; Zhang, Z.; Gou, L.; Wen, J.; Dong, J.; Wang, T., A Lipid-Anchored NAC Transcription Factor Is Translocated into the Nucleus and Activates Glyoxalase I Expression during Drought Stress. Plant Cell 2017,29, (7), 1748-1772.
  17. Mussig, C.; Schroder, F.; Usadel, B.; Lisso, J., Structure and putative function of NFX1-like proteins in plants. Plant Biol (Stuttg) 2010,12, (3), 381-94.
  18. He, F.; Niu, M. X.; Feng, C. H.; Li, H. G.; Su, Y.; Su, W. L.; Pang, H.; Yang, Y.; Yu, X.; Wang, H. L.; Wang, J.; Liu, C.; Yin, W.; Xia, X., PeSTZ1 confers salt stress tolerance by scavenging the accumulation of ROS through regulating the expression of PeZAT12 and PeAPX2 in Populus. Tree Physiol 2020.

Round 2

Reviewer 2 Report

This manuscript was clearly not presented in the identification of molecular characteristics of NF-X1 genes by analysis of bioinformatics.

See the lines of 109~115 in the manuscript, among the 8 NF-X1 genes have variable amino acids (from 358 amino acids of PtrNF-X1-1 to 1188 amino acids of AtNF-X1-1). but PtrNF-X1-2 and AtNF-X1-2 have high homology and then the sequence homology was directly related with their biological functions.  How have the sequence identity and similarity between both?

See the lines of 145~148, phylogenic analysis from Nf-X1 proteins were clustered in two branches. PtrNF-X1-3, PtrX1-4, AtNF-X1-1, and OsNF-X1-2 were belonged to the same class I or class II ? Correct to  class II. While PtrNF-X1-1, OsNF-X1-1, AtNF-X1-2 and PtrNFX1-2  clustered in class II. Correect to class I .

Is MeJa mean to methyl jasmonate?  Correct to JA to MeJa .

There is  no  rules in the expression profiles from four  PtrNF-X1 genes  during devlopment of polpar and stress. The manuscript was not suggested in the title by  the analysis of bioinformatics and  q-PCR.